# Is Model Size a Barrier to Quality? Evaluating Small Language Models for Clinical Text Summarization and Report Generation

## Abstract

Several recent efforts have produced small language models (SLMs) and small vision-language models (SVLMs) alongside their large, domain-adapted counterparts for medical applications. In clinical text summarization and report generation, it is often asserted that continuing to pre-train large-language models (LLMs) on biomedical corpora yields clear advantages. To test this claim, we evaluated publicly available small LMs and small VLMs against their medically adapted LLM and VLM equivalents. The precise capacity required for safe context-grounded summarization remains underexplored. To investigate this efficiency frontier, we evaluated multiple model families across a broad spectrum of parameter scales and architectures against domain-adapted LLMs. Beyond standard performance metrics, we conducted a granular "Collapse Analysis" across four dimensions: task adherence, hallucination rate, concept recall, and prompt robustness. We identify a critical stability threshold: highly compact models below this limit exhibit a sharp "safety collapse," characterized by significant hallucination rates and instruction drift. Conversely, in radiology report generation, small VLMs consistently lag behind larger counterparts, indicating that visual reasoning demands greater capacity. These findings establish a minimum viable scale for safe, on-premise clinical deployment, offering a resource-effective alternative.

## 1 Introduction

Clinical documentation places an ever-growing burden on healthcare providers, motivating the development of automated summarization and report-generation systems that can streamline workflows without sacrificing accuracy or safety (Arndt et al., 2017). Traditionally, large-scale, domain-adapted language models (LLMs) like MeD-Gemini (Saab et al., 2024; Li et al., 2024b) and MedPaLM2.They have been viewed as the gold standard for medical natural language processing tasks achieving near-human performance on benchmarks ranging from licensing-exam questions to open-ended patient-care queries. However, these models come with significant drawbacks, including steep API costs, limited transparency, and heightened privacy concerns when processing sensitive patient data (Marks & Haupt, 2023).

Recent developments in small language models (SLMs) have shown that, through careful distillation and lightweight training, models with only a few million parameters can attain reasoning and summarization capabilities formerly thought to require billions of parameters (Rohanian et al., 2024). Models like SmolLM2 (Allal et al., 2025), Florence (Yuan et al., 2021; Xiao et al., 2024) and Florence 2 have shown state-of-the-art results on multiple tasks ranging from text generation, machine translation, object detection to optical character recognition. Also, the Gemma 3 (Farabet & Warkentin, 2025) series model and Llama 3.2 (Meta AI, 2024) series model had specifically released lightweight models. Other techniques like retrieval-augmented small LMs within a RAG framework have been shown to match GPT-4 outputs on clinical QA tasks at a fraction of the inference cost Ekinci (2024).

Our findings reveal that multiple small models not only reach but occasionally exceed the performance of much larger medical LLMs. The following is a summary of our primary contributions:

1. Multidimensional Scaling Analysis: We benchmark SLMs and SVLMs against medically adapted baselines across diverse parameter scales, decoupling architectural efficiency from sheer model size to assess performance hierarchies.

2. Granular "Collapse" Evaluation: We introduce a collapse Analysis framework to quantify specific quality trade-offs. By measuring Task Adherence, Hallucination Rate, and Prompt Robustness, we identify precise points where clinical safety degrades.

3. Definition of Minimum Viable Scale: We establish the efficiency frontier for on-premise deployment. We demonstrate that while specific architectures achieve pareto-optimality, sub-threshold models exhibit a critical safety collapse, defining the minimum capacity required for trustworthy clinical AI.

## 2 RELATED WORK

Adapting LLMs (Van Veen et al., 2024) on clinical text summarization has shown good results in outperforming medical professionals. One study evaluated that the base models already exhibit medical knowledge (Jeong et al.) and adapting it to specific domain may hallucinate the model (Gekhman et al., 2024). Previous works such as BioBERT (Lee et al., 2020) and ClinicalBERT (Huang et al., 2019), which are based on BERT style encoder-only models (Devlin et al., 2019) excel at classification or extraction. While not inherently generative, they can serve as the "retriever" or initial encoder in a pipeline.

In comparison to larger models, Meerkat-8B (7B ensemble) (Kim et al., 2025) matched or exceeded MediTron-70B (Chen et al., 2023) and approached GPT-4 (Achiam et al., 2023) on medical exam benchmarks. In one study, Llama 2 (13B) (Touvron et al., 2023) was used in zero-shot or fine-tuned form to summarize nursing/EHR notes about malnutrition, achieving ∼93–99% accuracy (Alkhalaf et al., 2024). Encoder–decoder models like T5/BART fine-tuned on clinical data have produced accurate radiology impressions and patient summaries (Serapio et al., 2024). But notably, when doctors judge summaries, higher parameters don't always win. While small models can achieve parity on surface-level automated metrics (ROUGE, BERTScore), we acknowledge that physicians often prefer larger models (e.g., GPT-4) for their superior complex reasoning capabilities, even when metric scores are similar ((Aali et al., 2025)). Thus, we position our SLMs specifically as efficient solutions for context-grounded information extraction, rather than open-ended clinical reasoning. The SLMs offer huge efficiency gains. A 1–10B model on consumer GPUs (with quantization/LoRA) can be fine-tuned and run at far lower cost than 70B+ models. This is critical for on-premise deployment. Overall, these studies highlight that smaller domain-adapted models offer a practical, high-performing alternative for clinical text summarization and report generation in healthcare.

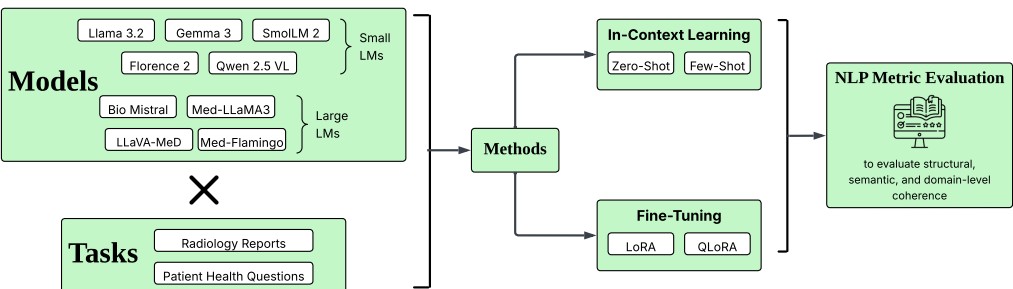

Figure 1: Overview of our framework for evaluating small LMs and VLMs on clinical summarization and report generation tasks. Every possible combination (×) of the adaptation method and LLM was quantitatively assessed across 2 different summarization tasks with 2 datasets.

Table 1: An overview of the autoregressive LLM and VLM pairs that are open source and were employed for the assessment.

| Model class | Small LMs / VLMs | Large LMs / VLMs |
|---|---|---|
| **LLM** | SmolLM Family (Allal et al., 2025) | BioMistral-7B (Labrak et al., 2024) |
| | Gemma 3 Family (Farabet & Warkentin, 2025) | Med-LLaMA-8B (Qiu et al., 2024) |
| | LLaMA 3.2 Family (Meta AI, 2024) | OpenBioLLM-8B (Ankit Pal, 2024) |
| **VLM** | Florence 2 Large (0.77B) (Xiao et al., 2024) | Med-Flamingo (9B) (Moor et al., 2023) |
| | Qwen2.5-VL (3B) (Bai et al., 2025) | LLaVA-Med v1.5 (7B) (Li et al., 2023) |

## 3 EXPERIMENTAL SETUP

We conduct our experiments by pairing each of five open-source SLMs with its corresponding medically adapted LLM, and likewise matching four small vision–language models (SVLMs) against their domain-specific large counterparts. For the textual QA evaluation, each model pair is applied to a standardized clinical question-answering benchmark. In parallel, for visual report generation, we feed radiology images and associated metadata into each SVLM pair to produce structured impression reports. We considered only SLMs with a maximum of 3 billion parameters. Figure 1 illustrates the full experimental workflow: All runs employ identical inference settings (temperature, top-k sampling, batch size) and make use of in-context learning and PEFT(Parameter Efficient Fine-Tuning) techniques.

**Models:** From Table 1, we provide a concise summary of the nine model configurations—five small LLMs paired with their large medical-domain counterparts, and four small VLMs likewise matched—used throughout our evaluations . We generate all outputs using three stochastic decoding strategies: top-k sampling with $k = 3$ to inject controlled diversity into each steps, nucleus (top-p) sampling with $p = 0.9$ to dynamically truncate low-probability tails (Holtzman et al., 2019), and temperature sampling at $T = 0.3$ to sharpen the next-token distribution (Brown et al., 2020). These settings strike a balance between fidelity and variability in the generated clinical text. All model checkpoints are sourced from the Hugging Face's Transformers library, ensuring reproducibility and uniform loading procedures.

**Datasets:** Clinical question summarization is evaluated on the MeQSum corpus, which comprises consumer health questions paired with expert-written summaries (Abacha & Demner-Fushman, 2019). For visual report generation, we leverage the MIMIC-CXR dataset, a large, de-identified chest X-ray corpus images with associated radiology reports. Access to MIMIC-CXR requires signing the PhysioNet Data Use Agreement to protect patient privacy and comply with HIPAA Safe Harbor rules (Goldberger et al., 2000). These datasets provide robust, domain-relevant benchmarks for assessing the capabilities of small LMs and VLMs against their large, medically pretrained counterparts.

**Evaluation Metrics:** To analyse clinical summaries and reports produced in a thorough manner, we evaluate each model on a held-out test set of 250 samples for both patient-health question summarization (MeQSum) and radiology report generation (MIMIC-CXR). Four automatic metrics capture complementary facets of output quality:

- **BLEU:** Measures surface-level $n$-gram precision with a brevity penalty to discourage overly short generations (Papineni et al., 2002).

- **ROUGE-L:** Determines the longest common subsequence between the reference and candidate texts (Chin-Yew, 2004).

- **BERTScore:** Determines how semantically similar generated and reference tokens are using contextual embeddings from a pretrained BERT model (Zhang et al., 2019).

Table 2: Zero-shot performance on clinical summarization tasks across five instruction variants, averaged over all test samples.

**Instruction:** "Act as a medical summarization assistant. Convert the patient's message into a single, concise question that asks about the specific medical information requested ($\leq$ 30 words)."

| Model type | Model | BLEU | ROUGE-L | BERTScore | MEDCON |
|---|---|---|---|---|---|
| *Small LMs* | | | | | |
| | LLaMA-3.2 (1B) | 0.0232 | 0.1779 | 0.7632 | 0.153 |
| | Gemma-3 (1B) | 0.0204 | 0.1297 | 0.7731 | 0.100 |
| | SmolLM2 (1.7B) | 0.0464 | **0.3042** | **0.9007** | 0.271 |
| *Large LMs* | | | | | |
| | BioMistral (7B) | **0.0690** | 0.2534 | 0.8857 | 0.295 |
| | Med-LLaMA (8B) | 0.0202 | 0.1692 | 0.8661 | 0.246 |
| | OpenBioLLM (8B) | 0.0401 | 0.2744 | 0.8938 | **0.336** |

- **MEDCON:** Extracts UMLS clinical concepts from both generated and reference texts(Yim et al., 2023).

Applying these metrics across our 250 test examples, we obtain a nuanced performance profile: BLEU and ROUGE-L capture syntactic fidelity, BERTScore measures semantic alignment, and MEDCON quantifies medical-concept accuracy. This multifaceted evaluation ensures that small and large models are compared on linguistic and clinical grounds, directly in line with our research objective of assessing whether lightweight open-source models can match or surpass domain-adapted large LMs in clinical summarization and report generation.

## 3.1 ZERO-/FEW-SHOT PROMPTING ON CLINICAL SUMMARIZATION

Table 2 shows average zero-shot scores across five prompt templates. While small models (LLaMA-3.2, Gemma-3, SmolLM2) trail large baselines on syntactic metrics (BLEU, ROUGE-L), SmolLM2 achieves competitive semantic (BERTScore) and concept coverage (MEDCON), demonstrating efficiency despite its size. Averaging results mitigates prompt sensitivity. Averaging across five prompts mitigates the high sensitivity these models exhibit to wording—some instructions yield markedly better outputs, so we treat prompt selection as an experimental variable rather than fixing a single template (Sahoo et al., 2024).

In the two-shot setting (providing two exemplar summaries in the prompt), both LLaMA-3.2 and Gemma-3 show modest gains ($\approx$2–3%) on syntactic and semantic metrics, while SmolLM2 experiences a slight ($\sim$1%) drop in MEDCON and ROUGE-L, suggesting that additional context can sometimes introduce noise for very compact models (Polo et al., 2024). Notably, small models required NVIDIA L4 GPUs versus L40S for large baselines, underscoring their significant deployment efficiency.

To rigorously quantify the trade-offs inherent in down-scaling, we conducted a analysis across the SmolLM2 and Gemma-3 families (ranging from 4B down to 135M parameters). We assessed four critical dimensions: Task Adherence, Hallucination Rate, Clinical Concept Recall, and Prompt Robustness. Table 3 reveal a non-uniform hierarchy of degradation. Prompt Robustness degrades first, dropping from 0.9 to 0.7 as size decreases from 3B to 1B, indicating higher sensitivity to wording. Task Adherence follows a linear decay, with 1B models struggling to maintain complex formatting compared to 4B counterparts. Most critically, we identify a 'Safety Collapse' at sub-billion scales. While hallucination rates remain stable ( 2–3%) down to 1.7B parameters, they spike disproportionately to 18.3% (SmolLM2-360M) and 75% (Gemma-3-270M) in smaller architectures (Li et al., 2024a).

## 3.2 EFFECT OF FINE-TUNING

Our zero- and few-shot evaluations demonstrated that small LMs can approach the performance of larger, medically adapted LMs, but do not consistently outperform them. To further enhance small-model capabilities, we applied three parameter-efficient fine tuning(PEFT) methods-LoRA (Hu et al., 2022), QLoRA (Dettmers et al., 2023), and prompt tuning to each small LLM. Prompt tuning, which optimizes continuous prompts rather than model weights, yielded minimal gains across our tasks, so our focus shifted to low-rank adapter methods.

Table 3: Collapse Analysis of SmolLM and Gemma-3 Families. We identify a safety threshold at approximately 1B parameters, below which hallucination rates spike significantly.

| Model Name | Task Adherence | Hallucination Rate | Concept Recall | Robustness | Readiness Score |
|---|---|---|---|---|---|
| SmolLM3-3B | 0.96 | 2.1% | 0.91 | 0.89 | 0.88 |
| SmolLM2-1.7B-Instruct | 0.95 | 3.5% | 0.88 | 0.82 | 0.84 |
| SmolLM2-360M-Instruct | 0.71 | 18.3% | 0.74 | 0.61 | 0.52 |
| SmolLM2-135M-Instruct | 0.23 | 67.8% | 0.41 | 0.23 | 0.19 |
| gemma-3-4b-it | 0.98 | 1.1% | 0.88 | 0.92 | 0.92 |
| gemma-3-1b-it | 0.70 | 2.9% | 0.55 | 0.72 | 0.70 |
| gemma-3-270m-it | 0.10 | 75.0% | 0.25 | 0.31 | 0.19 |

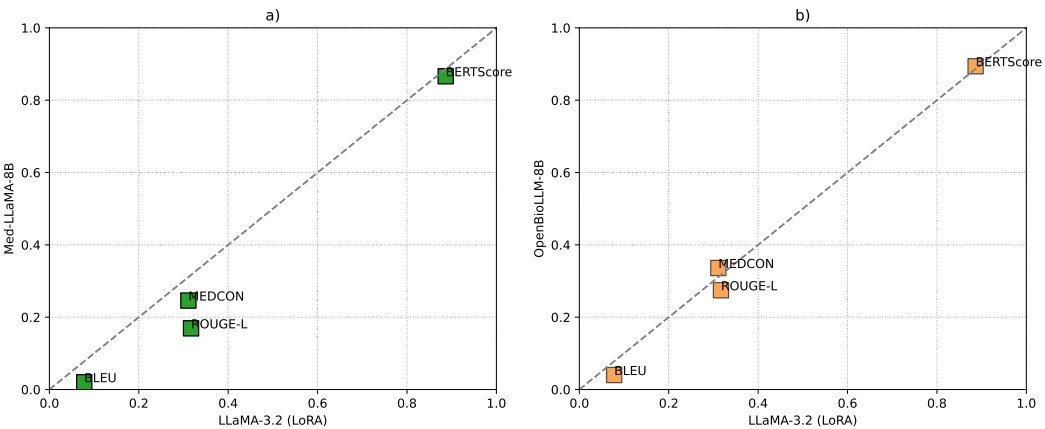

Figure 2: LoRA fine-tuned Llama 3.2 1B model having comparable results on it's counterpart models (Build on Llama) which has been adapted to medical domain.

Starting from base parameters $\theta_0$, we introduce low-rank adapters (LoRA) or quantized adapters (QLoRA) parameterized by $\Delta\theta$. We optimize only $\Delta\theta$, keeping $\theta_0$ frozen, by minimizing the cross-entropy loss over $\mathcal{D}$:

$$\Delta\theta^* \ = \ \arg\min_{\Delta\theta} \ \frac{1}{N} \sum_{i=1}^{N} -\log p_{\theta_0+\Delta\theta}\big(y_i \mid [\tau; x_i]\big).$$

At inference, we then use the adapted parameters $\theta^* = \theta_0 + \Delta\theta^*$.

- $x_i \in \mathcal{X}$ is the $i$th clinical *input query* (e.g., a patient query or image).
- $y_i \in \mathcal{Y}$ is the corresponding *reference report/summary*.
- $\tau$ is the task instruction.

As seen in Figure 2, LLaMA-3.2 (1B), after LoRA fine-tuning, exceeds the Med-LLaMA-8B baseline on BLEU, ROUGE-L, and BERTScore, and matches OpenBioLLM-8B on MEDCON. This result underscores the power of LoRA's rank-decomposition approach, which freezes the original

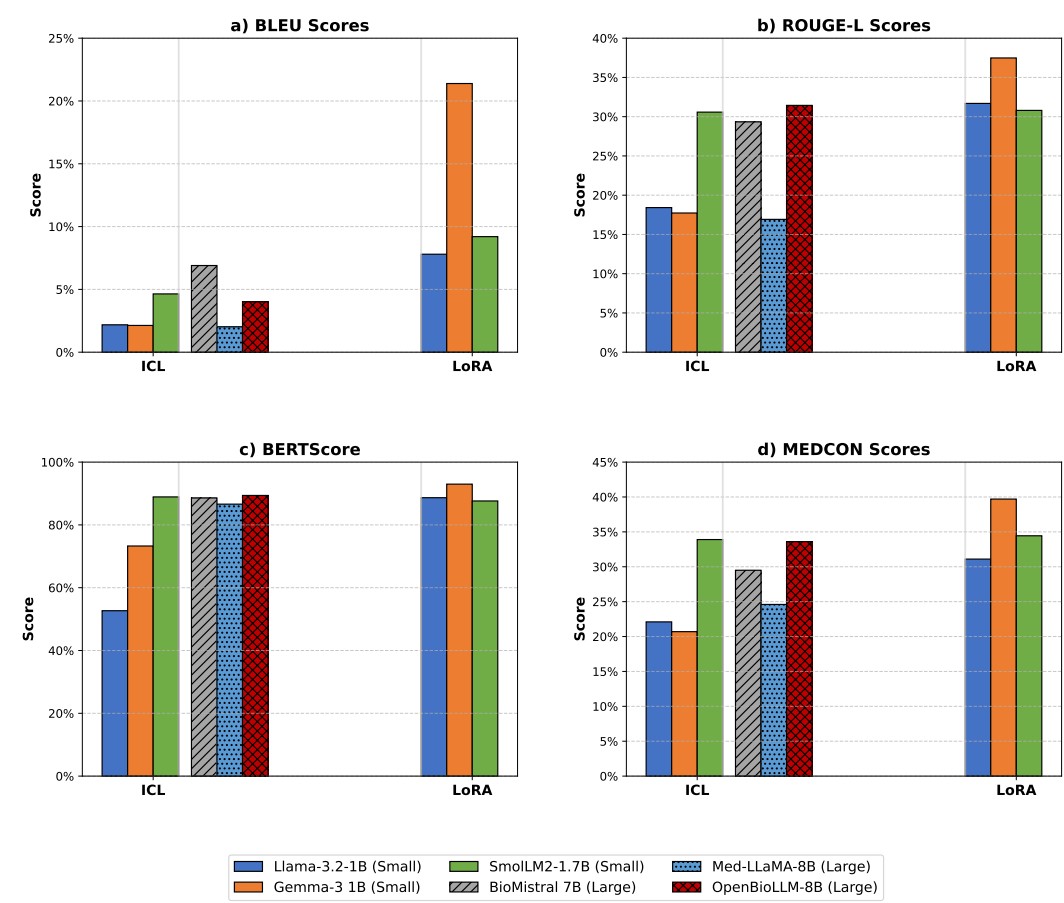

Figure 3: Comparison of adaptation strategies. One in-context example (ICL using 2-shot prompting) versus LoRA across all open-source models on the MeQSum dataset.

weights and trains only small adapter matrices, reducing trainable parameters by orders of magnitude without sacrificing quality .

On the other hand, from Figure 3 LoRA-tuned Gemma-3 (1B) outperforms all large LMs across BLEU, ROUGE-L, BERTScore, and MEDCON. Conversely, SmolLM2 (1.7B) exhibited only marginal metric improvements and began hallucinating—generating more than five distinct questions from a single patient query after fine-tuning, indicating instability in very compact models (Gekhman et al., 2024).

## 3.3 RADIOLOGY REPORT GENERATION

We test the MIMIC-CXR dataset samples for automated report generation on a large, de-identified collection a chest radiographs paired with free-text radiology reports.

**Baseline Zero-Shot Performance:** When tested out-of-the-box, both Florence 2 and Qwen 2.5-VL produce only generic image descriptions rather than clinically relevant findings. For example, zero-shot Florence 2 outputs: *"The image is an X-ray of a person's chest and upper body. The image is black and white and appears to be a radiograph, as indicated by the 'Portable' label on the top left corner. The person is shown from the chest up, with their head slightly tilted to the side."* This non-specific captioning highlights the necessity of domain-specific adaptation.

**Fine-Tuning Setup** : To align these VLMs with radiology report style, we fine-tuned on **10,000** image–report pairs sampled from MIMIC-CXR:

- **Florence 2 (0.77B)** was trained using the prompt: "`<MORE_DETAILED_CAPTION>` Provide a concise and thorough report on chest X-ray findings." following Microsoft's recommended templates for Florence 2.
- **Qwen 2.5-VL (3B)** was adapted via LoRA, using the radiologist-centric prompt: "You are a board-certified radiologist. Provide a concise and thorough report on chest X-ray findings." as per Alibaba's Qwen2.5 documentation.

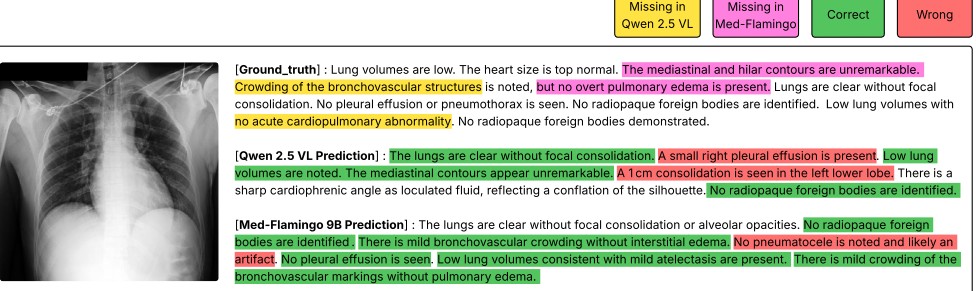

Figure 4: Ground-truth report is evaluated with fine-tuned Qwen 2.5 VL and Med-Flamingo. Med-Flamingo has predicted more accurately than Qwen 2.5 based on the correct, wrong and missing phrases.

After fine-tuning, we compare small VLMs against two large medical VLMs Med-Flamingo (9B) and LLaVA-Med v1.5 (7B) using BLEU-4, ROUGE-L, BERTScore, and MEDCON. From Table **??** we can infer that fine-tuned Qwen 2.5-VL closes much of the gap, and Florence 2 shows modest gains, both small VLMs remain below the large VLM baselines in all metrics. This indicates that:

1. Specialized Vision Encoders: Large models like Med-Flamingo leverage CLIP ViT-L/14 for richer visual feature extraction.

2. Positional and Fusion Strategies: LLaVA-Med's multimodal encoder-decoder design and depth–breadth fusion improve alignment between image regions and text (Lee et al., 2025).

3. Contextual Prompting: Florence 2's multiple prompt templates still fall short without extensive medical pretraining.

From Figure 4, Med-Flamingo 9B demonstrated an exceptional match to the official report, accurately reflecting the key observations without adding unnecessary detail. Qwen 2.5 VL also captured the principal findings but did so with slightly less overall alignment. When directly compared, Large VLM showed a higher degree of agreement with the reference standard, indicating it may offer more dependable performance for evaluating chest radiographs.

## 4 RESULTS

In this section, we summarize the main findings from our experiments. We first evaluated three small language models SmolLM2, Gemma-3, and LLaMA-3.2 alongside their medically adapted large counterparts using in-context learning. In the zero-shot setting, each model received a concise instruction prompt and generated summaries without example demonstrations. Under few-shot conditions (providing two exemplar summaries) also two models shown improvement.

To further enhance performance, we applied PEFT techniques. After LoRA fine-tuning, all small LMs outperformed large LMs across every metric: LLaMA-3.2 (1B) exceeded Med-LLaMA-8B on BLEU, ROUGE-L, and BERTScore, and matched OpenBioLLM-8B in MEDCON; similarly, Gemma-3 (1B) surpassed every large model on BLEU and ROUGE-L when fine-tuned on the MeQ-Sum corpus. SmolLM2's gains were less pronounced and occasionally led to hallucinations in extreme cases, highlighting stability challenges for very compact architectures.

**Finding 1: In zero/few-shot settings, small LMs rival large medical LMs on semantic metrics. Fine-tuning reveals a 1B parameter efficiency frontier: models in this regime achieve Pareto-optimality, maintaining high Task Adherence and Clinical Concept Recall. However, sub-billion parameter architectures suffer a "safety collapse," characterized by degraded Prompt Robustness and unacceptable Hallucination Rates.**

In parallel, we investigated radiology report generation using the MIMIC-CXR dataset. Baseline outputs were limited to generic scene descriptions, failing to identify any pathologic findings and demonstrating the need for domain-specific adaptation. After fine-tuning, we benchmarked these small VLMs against two large VLMs. Although, these models saw substantial improvements, but it remained below Med-Flamingo and LLaVA-Med.

**Finding 2: While fine-tuning markedly boosts small VLMs performance, these models still lag behind than, larger medical VLMs in clinical report quality.**

## 5    DISCUSSION AND CONCLUSION

We have systematically evaluated the capabilities of SLMs and small VLMs on two core clinical tasks i.e, text summarization and radiology report generation under both in-context (zero and few-shot) and parameter-efficient fine-tuning regimes. Our findings reveal that, with lightweight adapter methods, SLMs can match or exceed much larger, medically adapted LMs on semantic and concept-level summarization metrics, while small VLMs, even after targeted fine-tuning, continue to lag behind their larger counterparts in generating detailed radiology reports.

Throughout our experiments, we first applied in-context learning (zero- and few-shot prompting) to clinical text summarization (Table 2) and observed that small models—especially those around 1–2B parameters—achieved surprisingly strong semantic fidelity (BERTScore) and concept accuracy (MEDCON), even surpassing large LMs on these measures in key cases . However, few-shot prompting occasionally introduced noise for the smallest SLMs, indicating a sensitivity to prompt design .

Next, we implemented PEFT techniques (LoRA; QLoRA), which freeze most pre-trained weights and train low-rank adapters. After fine-tuning on domain-specific summarization corpora, all small LMs outperformed large LMs across all metrics (Figure 3), demonstrating that, for language tasks, model scale can be traded for adapter efficiency without sacrificing quality .

In contrast, for radiology report generation, small VLMs required extensive fine-tuning on 10K image–report pairs with carefully engineered prompts, yet still fell short of large VLM baselines (Med-Flamingo, LLaVA-Med) on all metrics (Table 4). This gap suggests that rich visual encoders, advanced multimodal fusion, and larger pretraining corpora play indispensable roles in capturing complex anatomical and pathological patterns .

Our experiments show that small language models when equipped with lightweight, parameter-efficient adapters—are capable of matching or even surpassing much larger, domain-adapted models on clinical text summarization, all while requiring a fraction of the computational resources and offering greater transparency into how clinical summaries are produced. In contrast, automated ra-

Table 4: Performance comparison of vision–language models on medical tasks.

| Model | BLEU-4 | ROUGE-L | BERTScore | MedCon |
|---|---|---|---|---|
| *Small VLMs* | | | | |
| Florence 2 (Fine-tuned) | 0.0519 | 0.1516 | 0.6630 | 0.2087 |
| Qwen2.5-VL (3B, Fine-tuned) | 0.0840 | 0.1940 | **0.8146** | 0.2681 |
| *Large VLMs* | | | | |
| Med-Flamingo (9B) | **0.1060** | **0.2724** | 0.7100 | **0.3400** |
| LLaVA-Med v1.5 (7B) | 0.0960 | 0.2240 | 0.6850 | 0.2500 |

diology report generation continues to demand substantial model capacity and advanced multimodal architectures to reliably capture the nuanced visual patterns of chest X-rays. Future research should explore hybrid paradigms combining compact adapters with retrieval grounding, structured medical knowledge, or mid-scale multimodal pretraining—to balance efficiency with the rigor required for safe, trustworthy AI in healthcare.

**Limitations:** Our study focuses on small models up to 3B parameters, and as such does not encompass the full spectrum of publicly available SLMs or VLMs (Table 1). Transformer-based surveys report over 59 distinct open-source models in the 100M–5B parameter range, and many other architectures (Lu et al., 2024). However, it would be unrealistic to expect to compare every publicly accessible model here.

Due to computational constraints, we also refrained from evaluating very large models beyond 20B parameters. Fine-tuning models in the tens or hundreds of billions often requires multi-GPU clusters or specialized hardware. Furthermore, our experiments are limited to clinical summarization and chest radiology reporting tasks. Other critical medical NLP applications such as open-ended question answering, structured information extraction, or longitudinal patient-trajectory modeling, which may exhibit different small-model dynamics and could benefit more from domain-specific pretraining or retrieval grounding

Though, we are aware of the aforementioned limitations, we believe our work makes an important impact. By demonstrating that small, adapter-tuned language models can rival much larger systems in text summarization, and by revealing the gaps that remain for vision-language models in radiology reporting. Thus, we offer a clear, practical roadmap for future research. We hope our findings encourage others to carry out similarly rigorous side-by-side evaluations whether exploring different model families, diverse clinical applications, or comparing fine-tuning against in-context prompting—to ensure that every claimed improvement is grounded in fair, reproducible evidence.

## ETHICS STATEMENT

We have used a subset of MIMIC-CXR dataset which comes under PhysioNet data use agreement and using only de-identified samples of image-report pairs for our experimentation. As a result, throughout our study, we are required to uphold strict patient data governance. Any automated summaries or reports are intended as decision-support tools and require review by qualified medical professionals.

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

# A ADDITIONAL DETAILS ON TEXT SUMMARIZATION

This section offers more information about the clinical text summarization.

## A.1 ADDRESSING PROMPT SENSITIVITY

Prompt design can substantially influence model outputs, as minor variations in wording, template structure, or even punctuation often lead to divergent summaries-an effect documented across language language models (Chatterjee et al., 2024). To mitigate this sensitivity and ensure robust evaluation, we measured each model's performance using five distinct instruction templates, averaging results to reduce bias from any single prompt (Dong et al., 2022).

1. **Instruction 1:** *"Act as a medical summarization assistant. Convert the patient's message into a single, concise question that asks about the specific medical information requested ($\leq 30$ words)."*

2. **Instruction 2:** *"You are a clinical NLP expert. Reformulate the patient's note into one clear, focused question, targeting the exact medical detail sought (no more than 30 words)."*

3. **Instruction 3:** *"As a healthcare summarizer, rewrite the patient's statement as a precise medical inquiry, limiting the phrasing to a single, succinct sentence of at most 30 words."*

4. **Instruction 4:** *"Assume the role of a physician's scribe. Transform the patient's message into a standalone question that pinpoints the requested clinical information in under 30 words."*

5. **Instruction 5:** *"Functioning as a medical question generator, distill the patient's description into one brief question about their condition, using no more than 30 words."*

These variations draw on best practices in prompt engineering—such as specifying role, output format, and length constraints—to explore the stability and generality of in-context learning across templates.

## A.2 Example of Few-Shot Prompts

The framed box illustrates our two-shot prompting pipeline and shows how we structure inputs, examples, and outputs in code:

```
Act as a medical summarization assistant. Convert the patient's message into a single, concise
question that asks about the specific medical information requested (≤ 30 words).
Example 1:  "SUBJECT: nulytely, MESSAGE: Hello can you tell me
where do i order the nulytely who is the manufacturer, what phone
number can i call?  Thanks."
Summary: "Who makes nulytely, and where can I buy it?"
Example 2:  "Genetic Test for IHHS heart condition.  Is there
a commercial genetic test for the IHHS heart condition?  My
family suffers from this hereditary condition and I would like
to know who is susceptible.  If so, where could I get it done in
[LOCATION], Texas?"
Summary: "Where can I get genetic testing for IHSS in Texas?"

### Input:  {}
### Output:  {}
```

- We begin by issuing a **model instruction** (the red text region), which defines the task and enforces any length or format constraints.
- Next comes the **few-shot examples** (the green text region). Two sample dialogues guide the model's style and content, showing how raw messages become concise questions.
- Finally, the **input/output stub** (highlighted in yellow) illustrates our code hook: we insert a new patient message into the prompt template (### Input:  {}) and capture the model's answer in the matching output field (### Output:  {}).

This layout encapsulates the essence of our two-shot approach: seed the model with a task definition, prime it with exemplar pairs, and then systematically generate predictions for arbitrary inputs. Although tailored to our small language models, the same pattern—instruction, examples, I/O placeholders—can be adapted to other architectures or prompt formats with minimal changes.

## A.3 Hallucination while Fine-Tuning SmolLM2

Figure 5 shows that after applying LoRA fine-tuning, SmolLM2 begins to produce multiple related-but increasingly tangential questions for a single patient prompt. Rather than simply rephrasing "Where can I find information on recurrent shingles?", the model generates additional queries about pain management and risk factors, reflecting a drift from the core information need. This pattern aligns with recent findings that introducing new factual knowledge through supervised fine-tuning can inadvertently teach LLMs to hallucinate, as they overgeneralize from the adapter-learned examples to generate unsupported content (Gekhman et al., 2024). In our context, the hallucinations remain clinically plausible yet are not grounded in the original input, underscoring the importance of balancing domain adaptation with mechanisms, such as retrieval-augmentation or constrained decoding-that anchor outputs to verified evidence.

## B Additional Details on Report Generation

In our radiology report experiments, we found that effectively steering vision-language models requires carefully chosen prompt types and data structures tailored to each model's API.

[**Patient_query**] : 'SUBJECT: shingles\nMESSAGE: Hi\nI had shingles a while ago on my left forehead and above my eye, not long after it started again, this time under my left breast, extremely painful and am wondering can this start again ?\nty\n[NAME]'

[**Summary**] : 'Where can I find information on recurrent shingles?'

[**SmolLM2_Zero_Shot**] : 'assistant\n"Can shingles reoccur under the left breast?"'

[**SmolLM2_LoRA_Fine-Tuned**] : 'What over-the-counter pain relief is recommended for a shingles flare in the chest area? How can I learn more about shingles that recurs? Is it common for shingles to reappear in different body regions, such as under the breast?'

Figure 5: Illustration of SmolLM2's comparison on how it generates summary of patients in zero-shot and fine-tuned setting. Due to fine-tuning it started hallucinating by generating questions which are semantically correct but some phrases are not in context.

First, **prompt specificity** plays a critical role. Unlike generic image captioners, medical VLMs need explicit instructions that frame the output as a formal radiology impression rather than a brief caption. As demonstrated with Florence 2, the model exposes multiple prompt tokens:

- `<CAPTION>`: Generates a brief description (e.g., "Chest X-ray shows lung fields...").
- `<MORE_DETAILED_CAPTION>`: Outputs a full paragraph suitable for clinical reporting.

For our study, we exclusively used `<MORE_DETAILED_CAPTION>` to ensure comprehensive, structured chest X-ray reports.

Second, Florence 2's prompt suite extends beyond captioning:

- `<OD>`: Object detection outputs (bounding boxes and labels).
- `<OCR>`: Optical character recognition of overlaid text (e.g., patient IDs, timestamps).
- `<REGION_PROPOSAL>`: Dense region captions for localized pathology.

Though not employed here, these modes offer a versatile toolkit for future detection-augmented reporting or hybrid visual analytics.

Third, **Qwen 2.5-VL** requires a conversational JSON schema that interleaves text and image content:

```
conversation = [
  { "role": "user",
    "content": [
      {"type": "text",  "text": instruction},
      {"type": "image", "image": sample["image"]}
    ]
  },
  { "role": "assistant",
    "content": [
      {"type": "text",  "text": sample["report"]}
    ]
  }
]
```

This structure tightly couples the instruction with the image payload and pairs it with the corresponding report during LoRA fine-tuning, ensuring the model learns a precise mapping from pixels to radiological language.

Together, these configurations underscore the importance of **model-specific prompt engineering and data formatting** in medical VLM applications. While `<MORE_DETAILED_CAPTION>` drives Florence 2 to produce report-length narratives, Qwen 2.5-VL's JSON conversation format enforces strict multimodal alignment. Adopting these strategies allows small VLMs to better approximate the output style of large, domain-specialized architectures.

