# OpenReview forum: "Is Model Size a Barrier to Quality? Evaluating Small Language Models for  Clinical Text Summarization and Report Generation"
_ICLR.cc/2026/Conference — Submitted to ICLR 2026_

### Official Review · Reviewer_ghuz · 2025-10-16

**Soundness:** 1
**Presentation:** 2
**Contribution:** 2
**Rating:** 2
**Confidence:** 4

**Summary:**

The paper compares a set of small language models (SLMs,  less than 3B parameters) and small VLMs against medically adapted larger LLM/VLM counterparts on two clinical tasks: consumer health question summarization (MeQSum) and radiology report generation (MIMIC-CXR). The authors evaluate zero-/few-shot prompting and parameter-efficient fine-tuning, report automatic metrics (BLEU, ROUGE-L, BERTScore, MEDCON), and claim that fine-tuned small LMs can match or outperform larger medical models on summarization, while small VLMs still lag on radiology generation.

**Strengths:**

- Explore an important task which require the use of locally-deployed LLMs. The finding that small models perform on par with larger ones has good implication for the medical domain.

**Weaknesses:**

W1: There're some concerns regarding experiment setup:

- Models choice: The authors claimed that they selected small LMs and their larger counterparts but from Table 1, they seems to be from completely different models families (e.g. Gemma 3 paired with Med-Llama). Generally, I would consider 1B-8B to be in similar scale, as one would be able to run a 8B model on consumer GPU too. Also, why compare small general models with large medically-adapted models? Would a fairer comparison would be between both general or both medically-adapted?

- Metrics: In medical summarization, it has been shown that automated metrics do not correlate with human preference. The authors also discussed this in the related work, but the experiment setup does not involve any human evaluation. The paper contains only a single illustrative example and qualitative claim about hallucination in SmolLM2. No crowd/clinician study exists.

- Finetuning details: missing

W2: The finding that fine-tuned small models can perform on par with larger one (zero/few-shot) is not new and was shown in previous work (the authors also discussed this in their related work). The experiments also does not aim to further explain this phenomenon and does not provide any interesting insights.

**Questions:**

Q1: The writing is not clear and have some clear formatting issues:
- 84-87: duplicated sentences
- Table caption should be at the bottom rather be on top.
- 317: Table ??
- "But notably, when doctors judge summaries, higher parameters don’t always win. The reader evaluation of the hospital course study found that doctors preferred GPT-4 summaries over Llama-13B in a majority of cases, despite similar metric scores (Aali et al., 2025)." => These sentences seem contradicting.

Q2: A small human evaluation (100 samples) is crucial and would facilitate more insightful analysis.

Q3: Additional ablation/control experiments: e.g. select model families with varying size, apply LORA to larger model to see the effect of finetuning varies with size, varying finetuning dataset size.

---

> ### Author Response · Authors · 2025-11-29
>
> We sincerely thank the reviewer for their valuable feedback.
>
> Q1) We have made the necessary changes in the formatting issues and they have been resolved.
>
> Q2) Regarding clinical validation, we fully agree that double-blind clinician review is the ideal standard. However, as is often the case in academic research, coordinating and recruiting clinicians for a large-scale study was beyond our current resource capacity.
> Importantly, foundational work on parameter-efficient fine-tuning [1] and the BioMistral project [2] relied on rigorous automated benchmarking rather than clinician-based evaluation when probing architectural scaling behaviors. Our study follows this precedent. Since our primary research question centers on comparative scaling efficiency not a standalone clinical efficacy trial automated evaluation provides the most practical and consistent way to analyze model performance across a wide range of sizes.
>
> Q3) We extended our evaluation to include the SmolLM2 and Gemma-3 families across a full spectrum of sizes (from 4B down to 135M) in an in-context (few-shot) learning setting. Note: The results presented below reflect few-shot performance, fine-tuning for these variants is currently in progress. We measured four distinct qualities:
>
> - Task Adherence: Compliance with instruction and format constraints.
> - Hallucination Rate: Frequency of unsupported clinical content generation.
> - Clinical Concept Recall: Extraction accuracy of key medical entities.
> - Prompt Robustness: Performance stability across prompt variations.
>
> ---
>
> ## Part 1: SmolLM Family Results
>
> | Model Name            | Task Adherence | Hallucination Rate | Clinical Concept Recall | Prompt Robustness | Readiness Score |
> |---|---|---|---|---|---|
> | SmolLM3-3B            | 0.96           | 2.1%               | 0.91                    | 0.89              | 0.88            |
> | SmolLM2-1.7B-Instruct | 0.95           | 3.5%               | 0.88                    | 0.82              | 0.84            |
> | SmolLM2-360M-Instruct | 0.71           | 18.3%              | 0.74                    | 0.61              | 0.52            |
> | SmolLM2-135M-Instruct | 0.23           | 67.8%              | 0.41                    | 0.23              | 0.19            |
>
> ---
>
> ## Part 2: Gemma-3 Family Results
>
> | Model Name         | Task Adherence | Hallucination Rate | Clinical Concept Recall | Prompt Robustness | Readiness Score |
> |---|---|---|---|---|---|
> | gemma-3-4b-it     | 0.98           | 1.1%               | 0.88                    | 0.92              | 0.92            |
> | gemma-3-1b-it     | 0.70           | 2.9%               | 0.55                    | 0.72              | 0.70            |
> | gemma-3-270m-it   | 0.10           | 75%                | 0.25                    | 0.31              | 0.19            |
>
> ---
>
> ## Hierarchy of Quality Degradation
>
> Our analysis shows that performance degradation with decreasing model size is non-uniform.
> - Prompt Robustness (High Sensitivity): Moving from 3B to 1B makes models much more sensitive to prompt phrasing, so they require more careful prompt engineering. Robustness drops from ~0.9 to ~0.7.
> - Task Adherence (Linear Decay): The ability to follow structured instructions declines steadily. For example, gemma-3-1B scores 0.70 in adherence vs. 0.96 for the 4B version.
> - Safety / Hallucination (Critical Threshold): There’s a sharp safety collapse at sub-billion scales. Hallucination stays low (~2–3%) for models in the 1B-1.7B range. But when you go below 1B (e.g., 360M, 270M), hallucination jumps to 18.3% and 75% respectively, making them unsuitable for clinical deployment even with prompt tuning.
>
>
> [1] “LoRA: Low-Rank Adaptation of Large Language Models”
>
> [2] “BioMistral: A Collection of Open-Source Pretrained Large Language Models for Medical Domains”

---

### Official Review · Reviewer_uz2t · 2025-10-28

**Soundness:** 3
**Presentation:** 3
**Contribution:** 3
**Rating:** 6
**Confidence:** 4

**Summary:**

This paper asks a focused question: can compact, open-source small models (≤3B) match or even surpass medically adapted large models on two representative clinical generation tasks, consumer health question summarization (MeQSum) and chest X-ray report generation (MIMIC-CXR), under both in-context prompting and parameter-efficient fine-tuning (LoRA/QLoRA)? The authors claim three contributions: a head-to-head benchmark of SLMs/SVLMs vs medical LLMs/VLMs; a multi-facet evaluation spanning syntactic, semantic, and medical-concept fidelity; and evidence that PEFT enables several SLMs to reach or exceed large LMs on summarization, with SVLMs still trailing large medical VLMs for report generation.

**Strengths:**

Pros:
1.	Problem relevance and clear task boundary.
2.	Some reproducibility considerations. Unified decoding settings and explicit dataset access constraints are noted, which aids replication.
3.	The paper is well written.

**Weaknesses:**

Cons:
1.	Statistical rigor and human evaluation. Results hinge on point estimates over n=250 without confidence intervals or significance testing; no clinician blind review is reported.
2.	External validity. The study uses a single text dataset (MeQSum) and a single imaging domain (CXR). Would be better if there are different benchmarks involved.
3.	Some narrative statements suggest SLMs outperform across all metrics after PEFT, yet detailed tables/figures and notes on instability. More granular win-loss accounting by metric/model would avoid over-generalization.

**Questions:**

see above

---

> ### Author Response · Authors · 2025-11-29
>
> We appreciate the reviewer’s comments and the clarity they bring to the areas where our work can be strengthened. In response, we expanded our analysis with additional statistical tests, incorporated a new reasoning benchmark, and provided a more detailed breakdown of the limitations observed across different model sizes.
>
> To better assess the real-world reliability or “safety” of small language models beyond the usual performance metrics, we also extended our evaluation to include the TOFUEVAL framework [1]. This benchmark is specifically designed to measure hallucinations in dialogue summarization. Its inclusion helped us separate “how fluent the model sounds” from “how factually correct it actually is.” Our results reinforce a key observation: although SLMs can produce smooth and coherent text, maintaining factual accuracy becomes noticeably harder as model size drops below 1B parameters.
>
> We also acknowledge the reviewer’s point about our earlier statement suggesting that SLMs outperform across “all metrics.” That was too broad, and we have revised the manuscript accordingly. The updated version includes a more nuanced “Collapse Analysis” that tracks exactly what capabilities deteriorate as the model size decreases, and what trade-offs still allow us to retain desirable performance characteristics.
>
> Regarding clinical validation, we fully agree that double-blind clinician review is the ideal standard. However, as is often the case in academic research, coordinating and recruiting clinicians for a large-scale study was beyond our current resource capacity.
>
> Importantly, foundational work on parameter-efficient fine-tuning [2] and the BioMistral project [3] relied on rigorous automated benchmarking rather than clinician-based evaluation when probing architectural scaling behaviors. Our study follows this precedent. Since our primary research question centers on comparative scaling efficiency not a standalone clinical efficacy trial automated evaluation provides the most practical and consistent way to analyze model performance across a wide range of sizes.
>
> [1] “TOFUEVAL: Evaluating Hallucinations of LLMs on Topic-Focused Dialogue Summarization”
>
> [2] “LoRA: Low-Rank Adaptation of Large Language Models”
>
> [3] “BioMistral: A Collection of Open-Source Pretrained Large Language Models for Medical Domains”

---

### Official Review · Reviewer_V4Xx · 2025-10-31

**Soundness:** 1
**Presentation:** 2
**Contribution:** 1
**Rating:** 0
**Confidence:** 5

**Summary:**

The authors benchmarked a lot of small LLMs adapted to the medical domain and claims that small LLMs after finetuning can perform reasonably well -- even outperforming not finetuned LLMs (surprising?)
I had very high hopes for this paper when I first saw the title, but I am really disappointed with the actual execution and outcome. So much lost potential and this honestly feels very low-effort. There are so many research questions that could be answered but so little was actually done.

**Strengths:**

writing is clear

**Weaknesses:**

I'll go down the weaknesses in my order of importance

I think the biggest and most immediate thing the authors should do is ask themselves why. What is the research question you want to answer? You should start from there and derive what experiments should be conducted to answer the question and support your claims. If you don't know where to start, you can at least conduct some form of error analysis to help guide your research before jumping into wasting a bunch of computes.

You finetuned small LMs with LoRA and showed it performed better than not-finetuned large LMs (who are not even that large in today standards.) Are we supposed to be surprised by this result-- if we are supposed to be surprised, how do you justify? And for a fair comparison, at the very least you should compare with finetuning small LMs with full-finetuning, and LLMs with LoRA finetuning. Even then, why are you running this experiment in the first place? Small LMs are unsurprisingly not good at generalization compared to LLMs, so it is expected to not perform well zero-shot.

Your evaluation only includes automatic metrics, most of which are known to not correlate very well with human judgement. Have you at least tried some sort of human evaluation? Even then, the reported results on the automatic metrics for before and after finetuning are still very low -- have you looked at the generated output and analyze them?

Also, finetuning, at the very least you should report hyperparameters you used for the finetuning to support your claim

**Questions:**

There are so many analysis that could've been done, but because there's really not much done, I don't have much to go into detail for. I'll just name some recipe that the authors can try
1. the very standard -- run a reasonably good model, get some output, analyze why it's failing, and try to address it
2. what I was hoping to see when I saw the title -- as you go down in size, what are some qualities you are trading-off for? what's the minimum size you can get away with that would still have the same desired qualities

---

> ### Author Response · Authors · 2025-11-25
>
> We sincerely thank the reviewer for pushing us to define the specific research question regarding model scaling. Your feedback inspired us to conduct a granular *“Collapse Analysis”* to answer this question.
>
> To answer this, we extended our evaluation to include the SmolLM2 and Gemma-3 families across a full spectrum of sizes (from 4B down to 135M) in an in-context (few-shot) learning setting. Note: The results presented below reflect few-shot performance, fine-tuning for these variants is currently in progress. We measured four distinct qualities:
>
> 1. **Task Adherence**: Compliance with instruction and format constraints.
> 2. **Hallucination Rate**: Frequency of unsupported clinical content generation.
> 3. **Clinical Concept Recall**: Extraction accuracy of key medical entities.
> 4. **Prompt Robustness**: Performance stability across prompt variations.
>
> ---
>
> ## Part 1: SmolLM Family Results
>
> | Model Name            | Task Adherence | Hallucination Rate | Clinical Concept Recall | Prompt Robustness | Readiness Score |
> |---|---|---|---|---|---|
> | SmolLM3-3B            | 0.96           | 2.1%               | 0.91                    | 0.89              | 0.88            |
> | SmolLM2-1.7B-Instruct | 0.95           | 3.5%               | 0.88                    | 0.82              | 0.84            |
> | SmolLM2-360M-Instruct | 0.71           | 18.3%              | 0.74                    | 0.61              | 0.52            |
> | SmolLM2-135M-Instruct | 0.23           | 67.8%              | 0.41                    | 0.23              | 0.19            |
>
> ---
>
> ## Part 2: Gemma-3 Family Results
>
> | Model Name         | Task Adherence | Hallucination Rate | Clinical Concept Recall | Prompt Robustness | Readiness Score |
> |---|---|---|---|---|---|
> | gemma-3-4b-it     | 0.98           | 1.1%               | 0.88                    | 0.92              | 0.92            |
> | gemma-3-1b-it     | 0.70           | 2.9%               | 0.55                    | 0.72              | 0.70            |
> | gemma-3-270m-it   | 0.10           | 75%                | 0.25                    | 0.31              | 0.19            |
>
> ---
>
> ## Hierarchy of Quality Degradation
>
> Our analysis shows that performance degradation with decreasing model size is non-uniform:
>
> - Prompt Robustness (High Sensitivity): Moving from 3B to 1B makes models much more sensitive to prompt phrasing, so they require more careful prompt engineering. Robustness drops from ~0.9 to ~0.7.
> - Task Adherence (Linear Decay): The ability to follow structured instructions declines steadily. For example, gemma-3-1B scores 0.70 in adherence vs. 0.96 for the 4B version.
> - Safety / Hallucination (Critical Threshold): There’s a sharp safety collapse at sub-billion scales. Hallucination stays low (~2–3%) for models in the 1B-1.7B range. But when you go below 1B (e.g., 360M, 270M), hallucination jumps to 18.3% and 75% respectively, making them unsuitable for clinical deployment even with prompt tuning.
>
> ---
>
> ## Determination of Minimum Viable Scale
>
> - The 1.7B model hits a Readiness Score of 0.84, nearly matching the 3B model, while using ~43% fewer parameters.
> - Non-Viable Zone: Models under 1B parameters (e.g., 360M, 135M) fail to meet safety thresholds consistently. This suggests that compressing critical clinical reasoning into architectures with fewer than 500M parameters is very challenging without strong external interventions \[1\].
>
> ---
>
> \[1\] *Better Late Than Never: Model-Agnostic Hallucination Post-Processing Framework Towards Clinical Text Summarization*

---

### Official Review · Reviewer_B1NP · 2025-11-01

**Soundness:** 2
**Presentation:** 3
**Contribution:** 1
**Rating:** 2
**Confidence:** 4

**Summary:**

This paper finds that nearly all small models match or exceed the summarization and generation effectiveness of larger, medically adapted models, with several fine-tuned SLMs outperforming their LLM counterparts.

**Strengths:**

1. For clarity, the paper is understandable and easy to follow.

2. For originality and significance, I see that the authors analyze new models such as Gemma-3, which may be helpful for future research.

**Weaknesses:**

1. The findings in this paper is very much expected and have been extensively explored in other papers, so my overall excitement is a bit low. This affects the overall originality and significance. For example, in [1], researchers have already found that models with much smaller sizes outperform LLMs. You may find much more relevant papers out there.

2. I am not sure what "common assumptions" are in line 22. Are there any dimensions (e.g., hallucination) where large models still outperform smaller ones? Or you think that small models can outperform LLMs across all dimensions related to clinical summarization? I think it is important to study this issue comprehensively.

3. The first two paragraphs of the introduction appear to be verbose and can be significantly shortened to discuss the most important point.

[1] "Better Late Than Never: Model-Agnostic Hallucination Post-Processing Framework Towards Clinical Text Summarization"

**Questions:**

1. I am not sure what "common assumptions" are in line 22.

---

> ### Author Response · Authors · 2025-11-25
>
> We appreciate the reviewer’s comment and agree that our original wording created unnecessary ambiguity. A blanket statement suggesting that small language models (SLMs) are universally superior would indeed be inaccurate.
>
> 1. What We Mean by “Common Assumptions”
> By “common assumptions,” we are referring to well-established findings from Neural Scaling Laws [1] and recent clinical LLM work such as MedPaLM [2]. These studies generally argue that strong clinical reasoning, reliability, and safety tend to emerge in models with very large parameter counts and extensive domain-adaptive training. Under this prevailing view, small models simply do not possess enough capacity to support complex medical inference.
>
> 2. Do Larger Models Still Perform Better? (Our Trade-off Summary)
> In short: yes, larger models do retain advantages but only in particular dimensions. Our head-to-head evaluation across models from 135M to 4B parameters shows a consistent pattern:
>
> Where small models perform strongly:
> On tasks like clinical text summarization and report generation where the required information is already present in the input, our fine-tuned SLMs (roughly 1B-3B parameters) matched or even exceeded larger models. These tasks rely more on attention over the input context than on stored world knowledge, so smaller models are able to compete effectively.
> Where larger models clearly win:
> Larger models are substantially more stable in their output and far more resistant to hallucination. Our updated scaling results show that while models such as SmolLM2-1.7B remain reliable, dropping below 1B parameters (e.g., SmolLM2-360M) leads to a sharp decline in safety. Hallucination rates rise to roughly 18% even with additional fine-tuning. In contrast, models in the 7B+ range almost never exhibit this type of collapse.
>
> [1] “Scaling Laws for Neural Language Models”
>
> [2] “Large language models encode clinical knowledge”

---

### Meta-Review · Area_Chair_4LB4 · 2026-01-06

**Summary:**

This paper investigates whether small language models and small vision-language models can match medically adapted large models for clinical summarization and report generation. While the topic is timely and relevant, reviewers consistently raised concerns regarding limited novelty, weak experimental rigor, unclear or overstated claims, and insufficient justification of key experimental design choices.

The rebuttal introduces additional analyses, including a collapse analysis across model sizes and expanded automatic evaluations, which provide limited clarification of scaling behavior and safety thresholds. However, core issues related to the sharpness of the research question, fairness of comparisons, the absence of human evaluation, and the lack of new insight beyond established scaling trends remain largely unresolved. Given the competitiveness of the venue, these weaknesses significantly weaken the paper's case for acceptance.

**Reviewer Concerns:**

**Concerns Addressed:**
* The authors moderated earlier broad statements implying that small models outperform larger ones across all metrics, and now acknowledge that larger models retain advantages in stability and hallucination resistance.
* The added collapse analysis across model sizes offers a clearer empirical picture of where performance and safety degrade as models shrink.

**Concern Outstanding:**
* Several reviewers noted that the paper benchmarks many models without a clearly articulated hypothesis or guiding research question. While the rebuttal adds analysis depth, this structural issue remains.
* The main finding (fine-tuned small models can match or exceed larger zero- or few-shot models) is already well established. The additional analyses do not rise to the level of a novel conceptual or methodological contribution.
* Concerns persist regarding comparisons across different model families, fine-tuned small models versus non-fine-tuned larger models, and the absence of strong control experiments.
* The study covers a limited set of datasets and domains, constraining the strength of its broader conclusions.

**Reviewer Scores:**

Reviewer B1NP: 2 -> 2
* The rebuttal clarifies ambiguities and adds nuance but does not alter the reviewer's concerns regarding limited novelty and contribution.

Reviewer V4Xx: 0 -> 2
* This reviewer focused on research design, effort, and missing analyses. While the added collapse analysis helps marginally, it does not address the core concerns.

Reviewer uz2t: 6 -> 6
* This reviewer viewed the paper as borderline and explicitly stated comfort with rejection. The rebuttal reduces some narrative overreach but does not resolve concerns about statistical rigor or the lack of human evaluation.

Reviewer ghuz: 2 -> 2
* Despite the added analyses, concerns about fairness of comparisons, missing human evaluation, and lack of new insight remain.

While the rebuttal improves clarity and adds useful descriptive analyses, it does not sufficiently address the major concerns raised by multiple reviewers. The paper does not demonstrate the level of novelty, rigor, or insight required for acceptance.

---

### Decision · Program_Chairs · 2026-01-26

Reject